# Yttria-Doped Ceria Surface Modification Layer via Atomic Layer Deposition for Low-Temperature Solid Oxide Fuel Cells

Hyeontaek Kim, Yongchan Park, Davin Jeong and Soonwook Hong *

Department of Mechanical Engineering, Chonnam National University, 77 Yongbong-ro, Buk-gu, Gwangju 61186, Republic of Korea
* Correspondence: shong@jnu.ac.kr; Tel.: +82-62-530-1686

**Abstract:** Atomic layer deposition (ALD) is performed to obtain less than 1 nm thick yttria-doped ceria (YDC) layers as cathode functional layers to increase the surface oxygen incorporation rate for low-temperature solid oxide fuel cells (LT-SOFCs). Introducing a YDC surface modification layer (SML) has revealed that the optimized yttria concentration in YDC can catalyze surface oxygen exchange kinetics at the interface between the electrolyte and cathode. The YDC SML-containing fuel cell performs 1.5 times better than the pristine fuel cell; the result is an increased exchange current density at the modified surface. Moreover, a heavily doped YDC SML degrades the performance of LT-SOFCs, owing to the weakened oxygen surface kinetics due to the increased migration energy of the oxygen ions.

**Keywords:** yttria-doped ceria (YDC); atomic layer deposition (ALD); surface modification layer (SML); solid oxide fuel cells (SOFCs); surface oxygen incorporation rate

## 1. Introduction

Solid oxide fuel cells (SOFCs) are considered promising renewable energy devices because they can directly convert chemical energy into electric energy with high energy conversion efficiency; in addition, they provide fuel flexibility and are pollution-free during operation [1–5]. However, conventional SOFCs typically require high working temperatures (800–1000 °C), which hinder their commercialization because of the thermal degradation of the catalysts and the narrow selection range of materials. Therefore, researchers have devoted much effort to lowering the operating temperature to values within the range in which the materials remain thermally stable (300–500 °C) [5–11]. Despite the lowered operating temperatures of SOFCs, the sluggish surface reaction rate of oxygen still degrades the performance. As the oxygen reduction reaction (ORR) between the cathode and electrolyte interface depends on the operating temperature, this behavior of low-temperature SOFCs (LT-SOFCs) is inevitable. In this regard, researchers have proposed to promote the ORR by employing a cathode functional layer (CFL). For example, thin films of highly conductive materials with oxygen ions have been used as CFLs to catalyze the surface oxygen reaction [11–15].

The most common electrolyte material for LT-SOFCs is yttria-stabilized zirconia (YSZ), which ensures an electrochemically stable performance. Nonetheless, the high activation energy (>1.0 eV) of YSZ hinders the use of this material as an electrolyte in the low-temperature range. Ceria-doped materials, which have low activation energies (<0.8 eV), are potential candidates for electrolytes [12–16]; for example, yttria-doped ceria (YDC) is a promising candidate because it has a higher ionic conductivity and faster oxygen exchange kinetics than YSZ in the low-temperature range [15,16]. The applications that have utilized yttria as a functional material can be found in [17–19]. Adding rare-earth cations such as $Y^{3+}$ to ceria can lead to lattice disorder and increase the oxygen vacancy concentration, rather than zirconia, following the mechanism of charge compensation. According to many

researchers, thin YDC films are suitable CFLs that promote the ORR kinetics between the cathode and electrolyte [15,16,20–24].

Researchers have fabricated CFLs via physical vapor deposition (PVD) and chemical vapor deposition (CVD). The former is usually used to create films that are more than a few tens of nanometers thick to cover the substrate surface completely. Thus, CFLs are generally fabricated with PVD (sputtering in particular) [25–27]. Atomic layer deposition (ALD), which is one of the finest technologies among the CVD methods, has been adopted to obtain extremely thin and dense films [28–37]. The ALD method can grow films based on a self-limiting reaction, which results in the precise control of the thickness at the atomic scale; this ensures the finest surface modifications for the desired compositions [28–30,35–37]. Moreover, the ALD process can easily control the thickness in nanoscale by repeating the super cycles, which consists of each precursor and oxidant pulsing cycles. Thus, ALD can be considered one of the strong candidates for the deposition of surface modification layers (SMLs) as CFLs for LT-SOFCs to increase the surface oxygen incorporation rate with less than 1 nm thick layers. Chao reported that excessive $Y^{3+}$ cation concentrations in SMLs can increase the oxygen exchange rate at the interface between the cathode and electrolyte by tuning the oxygen vacancy level at the surface [12]. This phenomenon implies that ALD can precisely manipulate the composition of SMLs on an atomic level by controlling the ratio between the host and dopant materials.

In this study, we modified a surface via ALD by controlling the ratio between yttria ($Y_2O_3$) and ceria ($CeO_2$) to enhance the surface oxygen reaction kinetics. One ALD super cycle is enough to catalyze the oxygen surface reaction with the resulting CFL, while more than 20% $Y_2O_3$ doping concentration in the YDC SML can maximize oxygen incorporation into vacancies. By contrast, excessive $Y_2O_3$ doping concentrations (more than 50%) in YDC SMLs decrease the oxygen exchange rate at the interface, owing to the higher migration energy of anions and binding energy between $Y_2O_3$ and oxygen vacancies. According to our results, an SML with an ALD cycle ratio of 2:5 ($Y_2O_3$-$CeO_2$) can maximize the surface oxygen exchange kinetics, resulting in a 1.5-fold increase in the peak power density compared to that of a pristine YSZ electrolyte fuel cell. The results of this study suggest that surface-modified YDC layers can successfully increase the surface oxygen exchange rate of LT-SOFCs. Therefore, the proposed surface modification technique based on ALD promotes catalytic activity and surface reactions for electrochemical applications.

## 2. Materials and Methods

### 2.1. Sample Preparation

To investigate the material properties and electrochemical behavior of YDC SMLs, two different types of substrates were used. We used an 8 mol% YSZ substrate (MTI Corporation) to fabricate the fuel cell, which has an area of 1 cm × 1 cm and 0.2 mm thickness with one polished side. In addition, a polycrystalline (100) silicon wafer (1 cm × 1 cm area with 0.5 mm thickness) was used to confirm the morphologies and compositions of the YDC thin layers. The different $Y_2O_3$-$CeO_2$ doping ratios for the single ALD super cycles for the fabrication of YDC SMLs are presented in Table 1.

**Table 1.** ALD deposition recipes for YDC samples. Each column shows one super cycle of the YDC fabrication process with different yttria–ceria deposition rates.

| Y1/Ce6 | Y2/Ce5 | Y3/Ce4 | Y4/Ce3 | Y5/Ce2 | Y6/Ce1 |
|--------|--------|--------|--------|--------|--------|
| Ce | Ce | Ce | Y | Y | Y |
| Ce | Ce | Y | Ce | Y | Y |
| Ce | Y | Ce | Y | Ce | Y |
| Y | Ce | Y | Ce | Y | Ce |
| Ce | Ce | Ce | Y | Y | Y |
| Ce | Y | Y | Ce | Ce | Y |
| Ce | Ce | Ce | Y | Y | Y |

The ALD process normally comprises a cycle for the precursor and oxidant, respectively. Ozone was used as an oxidant after each precursor for $Y_2O_3$ and $CeO_2$ pulsing. One super cycle of sequential ALD steps comprises seven cycles, which consist of $Y_2O_3$ and $CeO_2$ with different yttrium precursor pulsing rates converted 1 to 6 cycles. The metal precursors used for $Y_2O_3$ and $CeO_2$ deposition were tris(methylcyclopentadienyl)yttrium(III) (99.99%, Strem Chemicals) and tetrakis(2,2,6,6-tetramethyl-3,5-heptanedionato)cerium(IV) (minimum 97%, Strem Chemicals), respectively. Each precursor was heated up to 170 and 200 °C to ensure sufficient evaporation, respectively. The substrate was maintained at 250 °C during deposition. Argon (Ar) was introduced as a carrier gas at 0.8 Torr. The prepared YDC SMLs were mounted onto the cathode side of the YSZ substrates to fabricate fuel cells. Consequently, 80 nm thick porous platinum (Pt) films were deposited on both sides of the YSZ substrates as electrodes (i.e., the anode and cathode) via sputtering at 100 W DC power, 10 Pa working pressure, and room temperature. The whole surface of the YSZ substrate was covered with a Pt layer as the anode, and a 1 mm × 1 mm cathode was deposited onto the surface of the YDC SML with a mask.

Figure 1 shows a schematic of the ALD-fabricated YDC SMLs with different $Y_2O_3$ doping concentrations. In addition, the sequential ALD cycle is presented. Each ALD cycle consists of two steps: metal precursor pulsing and the introduction of the oxidant to remove the ligand from the metal precursor resulting from the formation of metal oxides (i.e., $Y_2O_3$ and $CeO_2$). Both steps must be followed by a purging step with Ar to evacuate the remaining precursor and oxidant. One super cycle comprises seven cycles with different $Y_2O_3$-$CeO_2$ ratios.

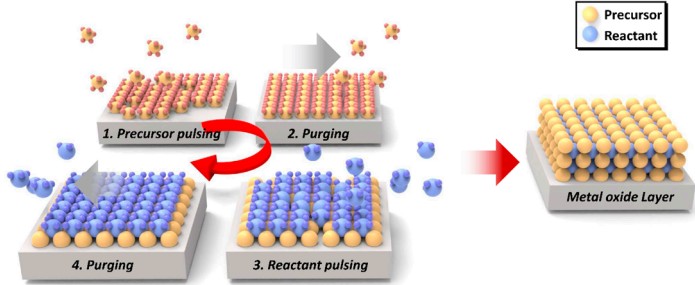

**Figure 1.** ALD process for YDC SML fabrication.

## 2.2. Fuel Cell Characterization

The YDC SML-coated silicon wafers were analyzed to determine the chemical compositions. X-ray photoelectron spectroscopy (XPS, Theta probe, Thermo Fisher Scientific Co., Waltham, MA, USA) and Al Kα monochromatic radiation were used to determine the molar ratios of the $Y_2O_3$ and $CeO_2$ contents. The film growth per ALD cycles were obtained from the film thickness, which is measured by variable angle spectroscopic ellipsometry (J.A. Wollam M2000 system).

The performance characteristics of the YDC SML-coated fuel cells were measured with a homemade probing system-containing temperature-controllable heating stage that provided a constant temperature (450 °C) during the fuel cell measurements. The current and voltage of the YDC SML-coated fuel cells were measured with a potentiostat (Gamry instruments Inc., Warminster, PA, USA, Reference 600). Additionally, the current and voltage of the YDC SML-coated fuel cells were measured by linear sweep voltammetry, changing the voltage from open circuit voltage (OCV, i.e., 1.0 V) to 0.2 V. The current density was calculated by dividing the measured current with the area of the cathode and then the power density was calculated by multiplying the voltage with the current density. Using the same apparatus, we conducted electrochemical impedance spectroscopy (EIS) between 1 MHz and 1 Hz. The equivalent circuit models of ZView (Scribner Association Inc., Southern Pines, N.C., USA) were used to analyze the acquired data. The electrochemical behavior of the sample was measured as depicted in Figure 2.

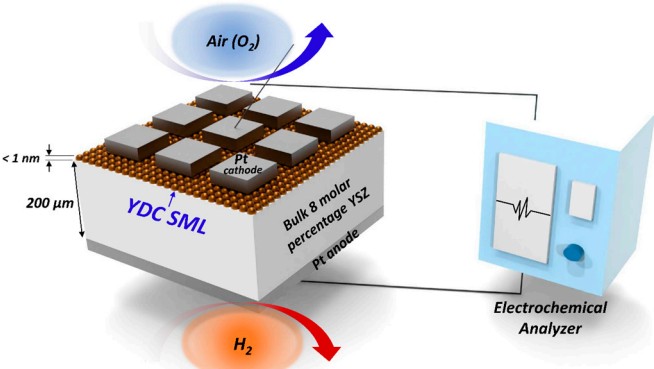

**Figure 2.** Schematic of YDC SML-coated fuel cell and electrochemical analysis system.

## 3. Results and Discussion

The chemical compositions of the different YDC SMLs were determined with XPS. Figure 3 shows the $Y_2O_3$ molar percentage as a function of the $Y_2O_3$ cycle number. The $Y_2O_3$ molar percentage was almost proportional to the $Y_2O_3$ cycle number. This phenomenon is reasonable because an increasing $Y_2O_3$ cycle number leads to a higher $Y_2O_3$ content during ALD. The $Y_2O_3$ concentration range in the YDC samples increased from 11.4 mol% to 79.8 mol% for the $Y_2O_3$ cycles 1–6. For more specific information of the chemical bonding state of the YDC layers, the representative XPS spectra for Y3d and Ce3d are shown in Figure 4. We confirmed that both yttrium and cerium were formed in the same oxidation state by observing the peaks' position and intensity.

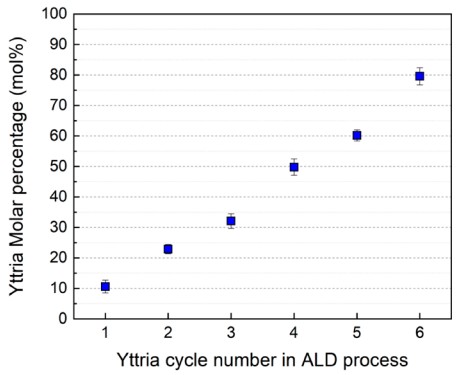

**Figure 3.** Relationship between yttria cycle number and yttria molar percentage in YDC SML.

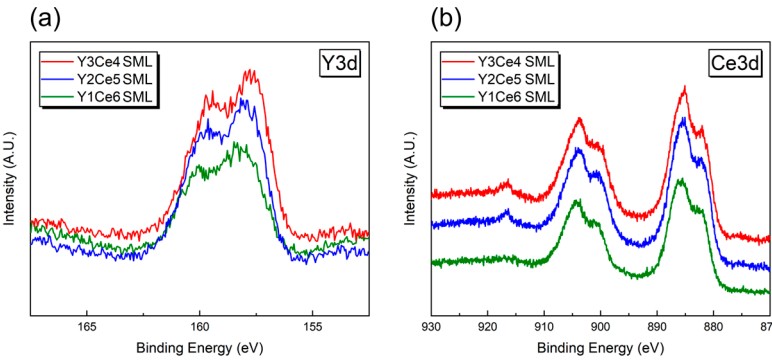

**Figure 4.** The representative XPS spectra for (**a**) yttrium and (**b**) cerium of YDC SML on the silicon wafer.

To estimate the YDC SML thickness, we conducted ellipsometry measurements. The $Y_2O_3$ and $CeO_2$ film thicknesses were measured by increasing the cycle number of the ALD

process (Figure 5). The growth rate of each element was calculated by dividing the thickness by the number of ALD cycles. According to the results, the growth rates of the ALD-fabricated $Y_2O_3$ and $CeO_2$ layers were 0.535 and 0.752 Å/cycle, respectively; these values agree well with those presented in earlier reports [15,16]. Nonetheless, the ellipsometry results for the ALD thin films that were deposited with fewer than ten cycles are not useful because the films may not fully cover the substrates. Therefore, we extrapolated the thicknesses of all the YDC SMLs from the ellipsometry results; they were less than 1 nm thick.

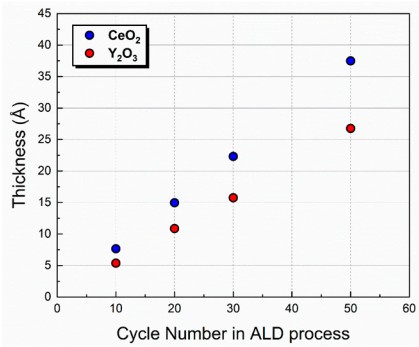

**Figure 5.** Thickness with respect to yttria and ceria cycle number in ALD process.

The performance characteristics of the SOFCs with different ALD recipes for the YDC SMLs and the bare YSZ electrolyte as a control sample were investigated by measuring the current-voltage (I-V) and current-power density (I-P) curves at 450 °C. As shown in Figure 6, the peak power density of the fuel cell with the bare YSZ layer was 4.2 mW/cm$^2$; those of the YDC SML-coated fuel cells with $Y_2O_3$-$CeO_2$ ratios of 1:6, 2:5, 3:4, 4:3, 5:2, and 6:1 were 5.5, 6.4, 6.1, 4.6, 3.6, and 2.6 mW/cm$^2$, respectively. The maximum power density increase was approximately 1.52, according to the values 6.4 and 4.2 mW/cm$^2$. The $Y_2O_3$ concentration of the best performing fuel cell with the YDC SML was 22.1 mol%, according to the XPS results (Figure 3). According to the previously published literature, YDC thin layers as CFLs show optimized oxygen ionic conductivity for 10–15 mol% $Y_2O_3$ [16,38]. These results reveal that $Y_2O_3$ doping concentrations above the optimized level may increase the oxygen vacancy density at the electrolyte surface and accelerate the incorporation of oxygen ions into the electrolyte. However, the performance was worse than that of the bare YSZ fuel cell when the YDC SML was subjected to more than five $Y_2O_3$ doping cycles. In other words, the performance was worse when more than 50 mol% $Y_2O_3$ was added to the YDC SML. Hence, excessive $Y_2O_3$ doping concentrations can increase the oxygen ion migration energy, which impedes the incorporation of surface oxygen and, thereby, affects the performance of the fuel cells.

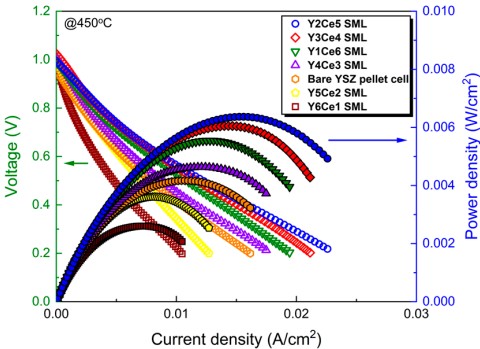

**Figure 6.** I-V-P curves of YDC SML-coated YSZ substrate-based fuel cells with different yttria concentrations measured at 450 °C. The curves which consist of the hollow marks indicate voltage-current density (I-V) and the filled marks represent power density-current density (I-P).

To identify the influence on the performance, EIS analysis was conducted at 450 °C and 0.8 DC bias voltage. The equivalent circuit model (depicted in Figure 7) was used to fit all of the EIS spectra to distinguish the effects of the first and second semicircles. The first semicircles of the impedance spectra in the high-frequency range meet the *x*-axis at 22.8 ± 0.4 $\Omega\cdot$cm$^2$. This resistance is generally associated with ionic transportation corresponding to ohmic resistance. We confirmed that the first semicircles of all the fuel cells showed similar characteristics; therefore, the effect of an increasing ohmic resistance can be negligible considering that the expected thickness of the YDC SML is less than 1 nm. However, the second semicircles of the impedance spectra, which are related to the electrode–electrolyte interfacial resistance (i.e., activation loss), showed different characteristics for different recipes. The YDC SML-coated fuel cell with a 2:5 $Y_2O_3$-$CeO_2$ ratio had a second semicircle, with a smaller radius; this indicates reduced interfacial resistance with a maximized surface oxygen reaction rate. In addition, the radii of the second semicircles of all the other fuel cells coated with different YDC SMLs were inversely proportional to the peak power density. Thus, a larger second semicircle radius leads to a smaller peak power density according to the EIS results. We confirmed that a YDC SML with an optimized $Y_2O_3$ concentration can improve the surface oxygen incorporation rate without increasing the ohmic loss.

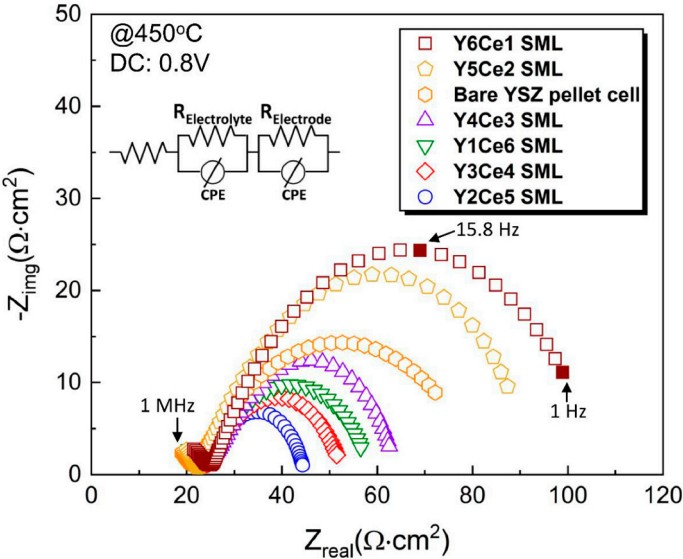

**Figure 7.** EIS spectra of YDC SML on YSZ substrates with different yttria concentrations.

The electrochemical characteristics of the different YDC SML-coated fuel cells are summarized in Figure 8. There was no significant difference among all the samples at an open circuit voltage above 1.0 V. However, the peak power densities of the fuel cells varied with the $Y_2O_3$ concentration, as shown in Figure 8a. To demonstrate which resistance affected the performance of the fuel cells, Figure 8b presents the area-specific resistance (ASR) associated with the ohmic and interfacial resistance (i.e., the polarization resistance) extracted from the EIS data. Evidently, the factor that deteriorates the fuel cell performance is mainly the ASR of the polarization rather than the ohmic resistance. Moreover, the optimized $Y_2O_3$ concentration of the YDC SML can considerably reduce the ASR of the polarization by enhancing the surface oxygen reaction kinetics, which improve the performance of the fuel cells.

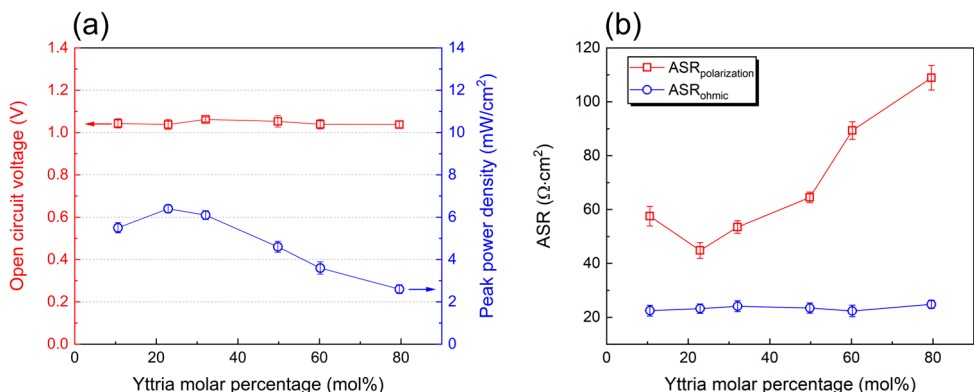

**Figure 8.** Electrochemical characteristics of (**a**) open circuit voltage and peak power density with respect to molar percentage of yttria; (**b**) area-specific resistance with respect to yttria concentration.

For a more thorough study of the surface oxygen exchange rate, we plotted the Tafel curves of the relationship between the activation overpotential ($\eta_{act}$) and natural logarithmic current density (ln $j$) for all the fuel cells (Figure 9). The value of $\eta_{act}$ can be derived from the theoretical open circuit voltage at 450 °C by subtracting the voltage measured in the experiments ($V_{meas}$) and ohmic potential component ($j \cdot ASR_{Ohm}$), which consists of the current density ($j$) and ASR from the ohmic resistance ($ASR_{Ohm}$):

$$\eta_{act} = OCV - V_{meas} - j \cdot ASR_{Ohm} \tag{1}$$

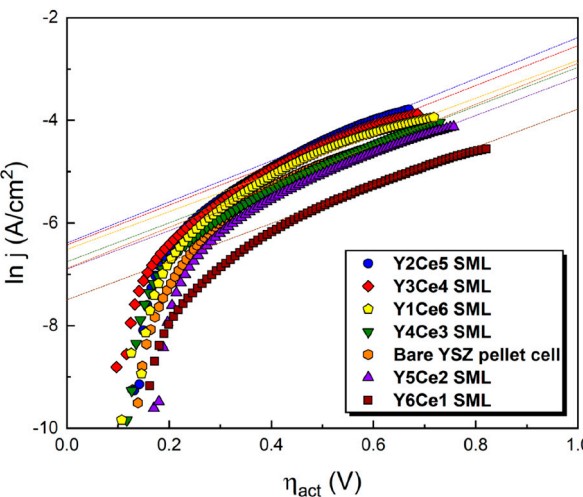

**Figure 9.** Tafel plots of YDC SML-coated YSZ substrate-based fuel cells.

When the current density $j$ is considerably greater than the exchange current density ($j_0$), the activation overpotential can be determined as follows:

$$\eta_{act} = -\frac{RT}{\alpha nF}\ln j_0 + \frac{RT}{\alpha nF}\ln j \tag{2}$$

where $R$ is the ideal gas constant, $T$ the absolute temperature, $\alpha$ the charge transfer coefficient, $n$ the number of electrons involved in the electrochemical reactions, and $F$ the Faraday constant. Regarding the $\eta_{act}$-ln $j$ Tafel plots, the y-intercept in the low overvoltage range provides the exchange current density $j_0$. The calculated $j_0$ values of the YDC SML-coated fuel cells with $Y_2O_3$-$CeO_2$ ratios of 1:6, 2:5, 3:4, 4:3, 5:2, and 6:1 were 1.39, 1.67, 1.57, 1.20, 0.99, and 0.54 mA/cm$^2$, respectively. The highest exchange current density was approximately 1.67 mA/cm$^2$ for the fuel cell with the 2:5 $Y_2O_3$-$CeO_2$ ratio, while the exchange current density of the bare YSZ fuel cell was 1.00 mW/cm$^2$. This indicates that the surface

oxygen incorporation rate with the optimized YDC SML is 1.67-fold greater than that of the YSZ electrolyte surface; this value agrees well with the 1.52-fold increase in the peak power density. Accordingly, the surface-modified YDC sample successfully improves the performance of the fuel cells by promoting the surface oxygen incorporation kinetics with the atomic-scale layer at the interface between the cathode and electrolyte. We expect that the SML concept based on ALD can be applied as a surface engineering approach in other catalyst applications.

## 4. Conclusions

Yttria-doped ceria (YDC), which is a promising alternative electrolyte material, was used as a functional layer in this study. The surface-modified YDC layer prepared with ALD improves the performance of LT-SOFCs. We introduced different $Y_2O_3$ concentrations into less than 1 nm thick YDC layers to tune the oxygen vacancy density at the interface between the cathode and electrolyte. According to the results of the electrochemical analysis, the $Y_2O_3$-$CeO_2$ ratio significantly affects the fuel cell performance. The optimal $Y_2O_3$ concentration was 22.1 mol%; this concentration can maximize the surface oxygen incorporation rate, whereas more than 50 mol% $Y_2O_3$ deteriorates the performance, owing to the increased migration energy of the oxygen ions. Consequently, a YDC SML that is fabricated with only one super cycle and with the optimized ALD recipe can accelerate the oxygen surface kinetics. We expect that the ALD-based SML concept will be applied to improve the catalytic activity in other electrochemical applications.

**Author Contributions:** This work was carried out in collaboration among all the authors. H.K. fabricated all the samples with ALD and analyzed the data of the electrochemical performance. Y.P. and D.J. analyzed the data of the XPS Ellipsometry. S.H. conceived, designed, and discussed this study. All authors have contributed, reviewed, and improved the manuscript. All authors have read and agreed to the published version of the manuscript.

**Funding:** This research was supported by "Regional Innovation Strategy (RIS)" through the National Research Foundation of Korea (NRF) funded by the Ministry of Education (MOE) (2022RIS-002) and the Basic Research Program through the National Research Foundation of Korea (NRF) funded by the MSIT (2022R1A4A3023960).

**Institutional Review Board Statement:** Not applicable.

**Informed Consent Statement:** Not applicable.

**Data Availability Statement:** Not applicable.

**Acknowledgments:** The authors would like to thank F.B. Prinz of Stanford University for their valuable advice and assistance in the experiments.

**Conflicts of Interest:** The authors declare no conflict of interest.

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
