# Peer review of "Yttria-Doped Ceria Surface Modification Layer via Atomic Layer Deposition for Low-Temperature Solid Oxide Fuel Cells"

_coatings, doi:10.3390/coatings13030491_

Round 1

Reviewer 1 Report

coatings-2224545

1.     Check reference  31. Also, please refer to some relevant recent work to keep the manuscript current. Please cite some recent MDPI references, also

2.     Y2O3 has several other applications, e.g. in security ink and PL etc. you should refer to them to make the intro more comprehensive (Synthesis and characterization of ultra-fine Y2O3: Eu3+ nanophosphors for luminescent security ink applications)

3.     Figure 7, please make the other y-axis coloured to mark which curve belongs to the first y-axis and which one to the second. Please explain the first increasing and then decreasing behaviour of the curve in figure 7 (a and b).

4.     In all the figures where bot Y-axis has been used, proper demarcations are missing

5.     What made you take the reading at 450 degrees celsius? 

Author Response

The authors appreciate the thoughtful comments, questions and suggestions provided by the reviewers.  All the response are described in attached file.

Reviewer 2 Report

1. The term "super cycle" should be explained in advance. (Probably in the Introduction section.

2. Table 1 and Figure 1 should be placed after the text.

3. Lines 121 to 128 are not the statement of results or discussion. Some of them should be in the section of Methods.

4.  Ellipsometry measurement method should be described in detail in the Methods section.

5. Measurement of current–voltage (I–V) and current–power density (I–P) curves should also be described further in the Methods section.

Author Response

(The authors gave the same response as above.)

Reviewer 3 Report

The manuscript entitled "Yttria-doped Ceria Surface Modification Layer via Atomic Layer Deposition for Low-Temperature Solid Oxide Fuel Cells" is devoted to the study of thin layers prepared by the ALD method for solid-state electrochemical devices. Despite the fact that the subject of paper is corresponds to the subject of "Coating" I cannot recommend the manuscript for publication (Reject). The work is weak, there are many questions, moreover, the investigated materials are no relevant for state-of-art solid-state devices.

- It is not clear why the authors make layers based on cerium and yttrium oxides on a YSZ support for low-temperature SOFCs. For a temperature of 450C, more reasonable to use dense layers of a supported electrolyte based on cerium oxide. Due to low temperatures, e-transport on fuel-side will be minimal.

- It is not specified how the electrodes were made. Sintering? Then at what temperature?

- As indicated in paper, Page.4, the thickness of one layer of CeO2 is 0.535 A, which is less than the unit cell parameter of cerium oxide (5.411 A), how is this possible?

- Was annealing performed after the deposition of oxides?

- How the overvoltage was measured?

- In paper is no proof of YDC film formation. In the conventional sense, a film implies a thin, continuous, uniform layer/coating of material on a substrate. Most likely, agglomerates or oxide particles are formed on the surface of the substrate.

- Describe the procedure for obtaining XPS spectra. At what angles were the XPS data obtained? Did you fix the matrix under the applied layers on the spectra? Give the spectra in the text of the manuscript.

- The information about ellipsometric measurements is absent in Materials and Methods.

Author Response

(The authors gave the same response as above.)

Round 2

Reviewer 1 Report

May be accepted

Reviewer 3 Report

The revised version of the manuscript looks better. Despite the fact that I still think that the work is weak, it can be accepted for publication.